# Advancing Genetic Tools in *Streptococcus pneumoniae*

**DOI:** 10.3390/genes11090965

**Published:** 2020-08-20

**Authors:** Haley Echlin, Jason W. Rosch

**Affiliations:** Department of Infectious Diseases, St. Jude Children’s Research Hospital, Memphis, TN 38105, USA; haley.echlin@stjude.org

**Keywords:** markerless cassette, *Streptococcus pneumoniae*, allelic replacement

## Abstract

*Streptococcus pneumoniae* is the causative agent of a multitude of diseases, and further study into its pathogenies is vital. The pneumococcus is genetically malleable, and several tools are available to manipulate this pathogen. In this study, we attempted to utilize one such tool, the Sweet Janus cassette, to replace the capsule locus with other capsule loci in our strain background and found that the efficiency of allelic replacement was low and the number of revertant false-positive colonies was high. We determined that the capacity to recombine capsule varied by the initial isolated colony, suggesting that frequency of reversion is dependent on the bacterial clone. Alternative selection markers may further expand the application of Sweet Janus. We created novel cassettes that utilized chlorinated phenylalanine as an alternative counter-selection agent in conjunction with the Janus or Sweet Janus cassette, providing a new dual or triple selection marker. Moreover, we created cassettes that do not require engineered resistance in the background strain, including both single and dual selection markers. We were able to utilize all constructs in allelic replacement of the capsule loci. These novel constructs provide a new means for generating gene deletions in *S. pneumoniae* that expand experimental applications.

## 1. Introduction

*Streptococcus pneumoniae* is a major human pathogen with high rates of mortality and disease, including sinusitis and otitis media, pneumonia, bacteremia, and meningitis [1]. Although a current vaccine against a subset of serotypes demonstrates limited coverage while providing great protection against colonization and invasive disease, the ability of the pneumococcus to exchange capsule, the vaccine target, and to readily obtain antibiotic resistance determinants underscores the importance for further study of this organism [2,3,4,5]. The role of any particular gene in pathogenesis can be readily assessed via genetic manipulation of *S. pneumoniae*, which is highly transformable [6]. While there are several means of creating genetic deletions, most involve introduction of resistant cassettes as a selectable marker to identify recombinants. Imparting resistance can have drawbacks in multiple forms, including inability to assess drug resistance in mutants due to potential cross-resistance, reduced number of potential combination of gene deletions, potential stress caused by production of resistance marker, and compromising experimental results. To overcome many of these shortcomings, a markerless cassette known as Janus was generated [7]. This cassette yielded the opportunity to study many gene functions in previously unavailable situations through allelic replacement or deletion [8,9,10].

The Janus cassette is comprised of a kanamycin (*kan*) resistance marker and a counter-selection marker that confers dominant streptomycin-sensitivity (*rpsL*). Allelic replacement occurs through a two-step transformation process. In the first step, the Janus cassette replaces the gene of interest through homologous recombination in a streptomycin resistant strain background, with selection for kanamycin resistance. With proper integration of the Janus cassette, mutants would be kanamycin resistant and streptomycin sensitive. In the second step transformation, the cassette is replaced with an alternate allele. These mutants would be kanamycin sensitive but regain streptomycin resistance upon loss of *rpsL* in the Janus cassette, which confers streptomycin sensitivity. One drawback to Janus is that the streptomycin-sensitive *rpsL* marker in the cassette can be spontaneously mutated to that of streptomycin resistance, leading to a high number of false-positive transformants that require a secondary screen [7]. To overcome this, a modified version of the Janus cassette was generated, termed Sweet Janus. This cassette follows the same principal of Janus but includes a second counter-selection marker (*sacB*) that confers sucrose sensitivity. With the Sweet Janus cassette, allelic replacement still occurs through a two-step transformation process. The first step is the same as with the Janus cassette, but upon the second step, the mutants would regain both streptomycin resistance and the ability to grow on sucrose. With two counter-selection markers, the revertant frequency and, thereby, number of false-positive transformants is greatly reduced [11].

Development of Sweet Janus has afforded a tool for more efficient allelic replacement in *S. pneumoniae* and has expanded the ability to study the impact of genetic disruption/replacement [12,13,14,15]. However, in this study, we have identified potential pitfalls. Successful recombination in the second transformation step was not equal in all bacterial clones. We found high variability in positive allelic replacement and false-positive frequency based on initial clone selection. This suggests that, for experiments that require multiple allelic replacements into the same background, a primary screening of Sweet Janus mutants would streamline downstream cloning. Additionally, both Janus and Sweet Janus rely on introduction of streptomycin resistance in the strain background prior to the first step of transformation. Thus, the continued presence of streptomycin resistance even upon allelic replacement of the cassette may limit the use of the mutated strain based on the desired downstream applications. As a potential alternative, we have constructed multiple novel cassettes that follow a similar two-step transformation protocol but no longer require generation of a streptomycin resistant strain and instead utilize sensitivity to growth on chlorinated phenylalanine as a selection marker. Moreover, we have created other cassettes that incorporate a combination of counter-selection agents of streptomycin resistance, sucrose sensitivity, and chlorinated phenylalanine sensitivity. While the efficacy of these new constructs was similar to Sweet Janus, these new tools will provide additional flexibility and experimental options for investigations with *S. pneumoniae*.

## 2. Materials and Methods

### 2.1. Growth Conditions

*Streptococcus pneumoniae* was inoculated from frozen stocks onto tryptic soy agar (Millipore Sigma, Billerica, MA, USA) plates supplemented with 3% sheep blood and 20 µg/mL neomycin and grown in 5% CO_2_ at 37 °C overnight. Bacterial culture on plates were used to inoculate liquid semi-defined media C + Y [16]. Strains used in this study are listed in Table 1 and the respective strain resistance and sensitivity are listed in Table 2.

### 2.2. Genomic DNA Extraction

Strains were grown in 10 mL C + Y to late logarithmic (OD_620_ ~0.8), and bacterial culture was pelleted at 6000× *g*, 10 min. For PCR (polymerase chain reaction) amplification and capsule swap experiments, genomic DNA (gDNA) was obtained using aqueous/organic extraction. Briefly, the bacterial pellet was subjected to lysis in 1 mL PBS plus 50 µL 10% SDS, 50 µL 10% DOC, and 10 µL of 10 mg/mL Proteinase K (Sigma, St. Louis, MO, USA), followed by incubation at 37 °C until clear. Lysates were mixed with 500 µL of phenol: chloroform: isoamyl alcohol (Sigma) and transferred to phase-lock tubes (Quantabio, Beverly, MA, USA). Organic and aqueous phases were separated by centrifugation as per instructions. The aqueous phase was mixed with 500 µL chloroform: isoamyl alcohol in the phase-lock tube, separated again by centrifugation, and then transferred to 100% ethanol for DNA precipitation. Precipitated DNA was washed in 70% ethanol, dried at 65 °C, and rehydrated in water. For confirmation of mutations, genomic DNA was extracted from pneumococcal strains using a modified version of the Wizard DNA extraction kit (Promega, Madison, WI, USA). The bacterial pellet was subjected to lysis in 500 µL PBS plus 50 µL 10% SDS and 50 µL 10% DOC, followed by incubation at 37 °C until clear. RNA was removed by addition of 3 µL of 4 mg/mL RNAseA (Promega) and incubation at 37 °C for 15 min. The remainder of the extraction followed the protocol provided with the kit, starting with step 3.

### 2.3. PCR Amplification and Transformation

All mutations were generated through transformation of *S. pneumoniae* with PCR products created through splicing by overlap extension (SOE) PCR (Appendix A) [17]. All PCR products were amplified using exTaq polymerase (TAKARA, Mountain View, CA, USA) following the recommended guidelines. Primers used are listed in Table 3. Template, primers, and expected size of each PCR product are listed in Table 4. To transform *S. pneumoniae*, PCR fragments were introduced to TIGR4 or D39 grown to OD_620_ ~0.07 in C + Y, along with competence stimulating peptide 2 or 1, respectively [18]. Cells were incubated for 3 h at 37 °C, and mutants were selected on TSA agar plates containing 3% sheep blood and 20 µg/mL neomycin plus the selective agent (Table 5): 800 µg/mL streptomycin (Sigma), 400 µg/mL kanamycin (Fisher, Fair Lawn, NJ, USA), 10% sucrose (Sigma), or 7.5–15 mM chlorinated phenylalanine (Cayman # 26168, Ann Arbor, MI, USA) [11,19,20].

### 2.4. Generation of Streptomycin Resistant Strains

In this study, we aimed to delete the capsule locus with the Sweet Janus cassette, followed by allelic replacement with other capsule loci. Because the selection for allelic replacement requires the background strain to have streptomycin resistance, we first introduced this resistance to our TIGR4 and D39 strains. While there are multiple means of conferring streptomycin resistance, we mutated the gene encoding the 30 s ribosomal subunit S12, *rpsL* (Sp_0271; Spd_0251) via point mutation at nucleotide 167 (AAA→ACA), resulting in amino acid change K56T. We followed this strategy because it affords a known source of resistance, limiting the potential secondary mutations arising from spontaneous selection, and because it yields high levels of resistance [19]. The point mutation in *rpsL* was generated through SOE PCR using two fragments amplified from TIGR4 genomic DNA (Appendix A). The upstream fragment (PCR A) consisted of the region about 1 kb upstream to 15 bp downstream of the point mutation and was amplified using primers HE01/HE02. The downstream fragment (PCR B) consisted of the region 15 bp upstream to about 1 kb downstream of the point mutation and was amplified using primers HE03/HE04. To generate the point mutation, HE02 and HE03 were designed so that the middle nucleotide of the primer was mutated. The mutated SOE PCR (PCR 1) was amplified using PCR A and B as templates and primers HE01/HE04 and was introduced into TIGR4 and D39 via transformation with selection of 800 µg/mL streptomycin—generating TIGR4S and D39S. Presence of the point mutation was confirmed in the SOE PCR and in the amplified gene from the chromosome of mutated strains via sequencing with primer HE05. No other mutations were observed in *rpsL*.

### 2.5. Replacement of Capsule Locus with Sweet Janus and NewSweet Janus Cassette

After confirming that TIGR4S obtained resistance to streptomycin, we replaced the TIGR4 capsule locus with Sweet Janus as described previously to create strain TIGR4SΔcps::SweetJanus [11]. Briefly, we amplified the region from *dexB* (Sp_0342) through *aliA* (Sp_0366) from SpnYL001 (generously provided by Yuan Li from the Harvard School of Public Health, Cambridge, MA, USA), which incorporates the Sweet Janus replacement of the capsule locus (Appendix A). This DexB-SweetJanus-AliA fragment (PCR C) was amplified using primers HE06/HE07, which have the same sequence as the primer pair (YL229/YL236) used to generate the allelic replacement previously [11]. This fragment includes genes that confer sucrose sensitivity (*sacB*), kanamycin resistance (*kan*), and streptomycin sensitivity (*rpsL*). The fragment was introduced into TIGR4S and mutants were selected with 400 µg/mL kanamycin. We isolated colonies that had no resistance to the counterselection agents, 800 µg/mL streptomycin plus 10% sucrose. If colonies did have resistance, they were considered to not be mutants. Mutation was confirmed by loss of capsule via agglutination using latex beads bound to antisera against type 4 capsule (Statens). D39SΔcps::SweetJanus was generated in a similar manner by transformation of D39S with PCR C.

To extend homology available for recombination, we created the modified “New” Sweet Janus cassette by splicing together three fragments using SOE PCR (Appendix A). The first fragment (PCR D) was the entire *dexB* gene with an overhang sequence complementary to *sacB* plus the promoter region upstream of *sacB* from Sweet Janus cassette. This fragment was amplified from TIGR4 gDNA using primers HE08/HE09. The second fragment (PCR E) was comprised of the promoter region upstream of *sacB*, *sacB*, *kan*, and *rpsL* and was amplified from SpnYL001 gDNA using primers HE10/HE11 based on the Sweet Janus sequence (Accession# KJ845726). The third fragment (PCR F) was the entire *aliA* gene with an overhang sequence complementary to *rpsL* and was amplified from TIGR4 using HE12/HE13. Amplification using these primers only amplified *aliA* plus probable native promoter and excludes the upstream region included in the original DexB-SweetJanus-AliA, as this region has homology to D39, but not to TIGR4 (Appendix A). The SOE PCR (PCR 2) was amplified using PCR D, E, and F as templates and primer pair HE08/HE13. TIGR4SΔcps::NewSweetJanus was generated by transformation of TIGR4S with PCR 2, with selection of 400 µg/mL kanamycin. Mutants were confirmed to be unencapsulated via latex agglutination and to have no resistance to counterselection agent, 800 µg/mL streptomycin plus 10% sucrose. D39SΔcps::NewSweetJanus was generated in a similar manner by transformation of D39S with PCR 2.

### 2.6. Allelic Replacement Efficacy

To determine the efficacy of allelic replacement in SpnYL001, TIGR4SΔcps::SweetJanus, and TIGR4SΔcps::NewSweetJanus, we determined the number of isolated colonies that recombined a type 2 capsule (true-positive) and the number of colonies that failed to recombine (false-positive) upon transformation with 4 µg of D39 gDNA. A colony was deemed to be true-positive if it was smooth (mucoid and opaque), while a colony was deemed to be false-positive if it was rough (non-mucoid and dull) and resembled the pre-transformed unencapsulated strain. A subset of true-positive and false-positive colonies was confirmed by latex agglutination for each transformation. We followed the same guidelines to test for variability in transformation efficacy among isolated colonies of TIGR4SΔcps::NewSweetJanus. The isolated colonies of TIGR4SΔcps::NewSweetJanus were obtained by selecting eight colonies from the transformation generating TIGR4S by introduction of the mutated *rpsL* (PCR 1) into TIGR4, followed by selection of two isolated colonies from each transformation generating TIGR4SΔcps::NewSweetJanus by introduction of PCR 2 into the eight TIGR4S colonies. Similarly, to determine the efficacy of allelic replacement of specific capsule loci, we transformed TIGR4SΔcps::NewSweetJanus or D39SΔcps::NewSweetJanus via introduction of 4 µg of gDNA extracted from CDC isolates of different serotypes. We determined the number of isolated colonies that recombined the capsule loci (true-positive) and the number of colonies that failed to recombine (false-positive). Colonies that were smooth (mucoid and opaque) were deemed true-positive, and colonies that were rough (non-mucoid and dull) were deemed false-positive, with a subset confirmed by latex agglutination.

### 2.7. Allelic Replacement Efficacy of spxB

To determine if the observed variances of transformation efficacy occurs upon allelic replacement of other loci other than capsule, we performed a two-step transformation to replace *spxB* (Sp_0730) in TIGR4S, D39S, and in two clinical backgrounds—ABCA31S and CDC001S. To utilize Sweet Janus, streptomycin resistance was conferred first, as described above—by transformation of TIGR4, D39, ABCA31, and CDC001 with PCR 1 and selection of mutants on 800 µg/mL streptomycin—Generating TIGR4S, D39S, ABCA31S, and CDC001S. Two colonies from each transformation were selected for allelic replacement of *spxB*. In the first step transformation, *spxB* was replaced by Sweet Janus using SOE PCR. The upstream fragment (996 bp upstream of the start codon) was amplified using HE25/26 and the downstream fragment (962 bp downstream from the stop codon) was amplified using HE27/HE28, both from TIGR4 gDNA. Sweet Janus was amplified using HE10/HE11 from SpnYL001 gDNA. The SOE PCR was amplified using the upstream, downstream, and Sweet Janus fragments as template and primer pair HE25/HE28. The spxB::SweetJanus SOE PCR was introduced into the two colonies of TIGR4S, D39S, ABCA31S, and CDC001S and mutants were selected with 400 µg/mL kanamycin. Four colonies of ΔspxB::SweetJanus from each transformation were selected for the second step transformation (except for ABCA31S, where one and seven colonies were selected from the two ABCA31S colonies). For the second step transformation, Sweet Janus was replaced with a cassette conferring erythromycin resistance (Erm) by amplification of ΔspxB::Erm PCR from TIGR4ΔspxB::Erm using HE25/HE28. This PCR was introduced into the eight isolated colonies of TIGR4SΔspxB::SweetJanus, D39SΔspxB::SweetJanus, ABCA31SΔspxB::SweetJanus, and CDC001SΔspxB::SweetJanus, and transformants were selected on plates containing either 800 µg/mL streptomycin plus 10% sucrose or 1 µg/mL erythromycin. Colonies that grew on erythromycin were considered to be true-positive as Sweet Janus was replaced with Erm. Colonies that grew on streptomycin plus sucrose consisted of both true-positive (loss of Sweet Janus) and false-positive (loss of streptomycin sensitivity) colonies. The number of false-positive colonies was calculated by subtracting the number of true-positive colonies from the total number of transformants (those that grew on streptomycin plus sucrose).

### 2.8. Construction of Novel Markerless Cassettes for Allelic Replacement of Capsule Locus

To provide alternative markerless cassettes based on chlorinated phenylalanine sensitivity, we first generated a mutated form of *pheS* (Sp_0579 and Spd_0504), which encodes the α unit of the phenylalanyl-tRNA synthetase. A point mutation in *pheS* leading to amino acid change A315G allows for integration of chlorinated phenylalanine into proteins. With recombination of the mutated *pheS*, the pneumococcus will struggle to grow on media containing chlorinated phenylalanine as seen in other streptococci and as predicted previously [20]. To mutate *pheS* to generate amino acid change A315G (Appendix A), two PCR fragments were amplified from TIGR4 gDNA. The upstream fragment (PCR G) consisted of the region from the start codon of *pheS* to 15 bp downstream of the point mutation (C to G at nucleotide 944) and was amplified using primers HE14/HE15. The downstream fragment (PCR H) consisted of the region 15 bp upstream of the point mutation to the stop codon of *pheS* and was amplified using primers HE16/HE17. To generate the point mutation, HE15 and HE16 were designed so that the middle nucleotide of the primer was mutated. The SOE PCR (PCR 3) with the point mutation was amplified using PCR G and H and primers HE14/HE17. Proper mutation of *pheS* was confirmed via sequencing with primer HE18.

The mutated *pheS* (PCR 3; *pheS* A315G) was incorporated into four novel cassettes: Phun Janus (*pheS* A315G, *rpsL*, *kan*), PhunSweet Janus (*pheS* A315G, *sacB, rpsL*, *kan*), PhunSweet (*pheS* A315G, *sacB*, *kan*), and Phun (*pheS* A315G, *kan*). As the novel cassettes were to be utilized for allelic replacement of the capsule locus, we generated Phun Janus, PhunSweet Janus, PhunSweet, and Phun by splicing the cassettes with DexB and AliA, the flanking genes of the capsule locus, via SOE PCR. Similar to Sweet Janus, Phun Janus relies on the counterselection of streptomycin sensitivity through *rpsL*, making it a necessity to first confer streptomycin resistance in the strain background. In contrast, PhunSweet and Phun do not contain *rpsL* and, thus, may be utilized in the native strain background. PhunSweet Janus has three possible counter-selection agents. In this study, a triple selection was utilized, requiring a strain background with streptomycin resistance. However, it may be possible to utilize only the chlorinated phenylalanine and the sucrose sensitivities of this cassette for counter-selection, allowing for use of this cassette in streptomycin sensitive strains.

For Phun Janus (Appendix A), the upstream fragment (PCR I) included *dexB* plus the promoter region upstream of *sacB* from the Sweet Janus cassette, with an overhang complementary to the 5′ end *pheS*. The downstream fragment (PCR J) consisted of *kan*, *rpsL*, and *aliA* from Sweet Janus, with an overhang complementary to the 3′ end of *pheS*. Both fragments were amplified from SpnYL001 gDNA using primers HE06/HE19 (upstream fragment) and primers HE20/HE07 (downstream fragment). The SOE PCR (PCR 4) was amplified using PCR I, PCR J, and *pheS* A315G (PCR 3) with primers HE06/HE07 and was introduced into TIGR4S and D39S to generate TIGR4SΔcps::PhunJanus and D39SΔcps::PhunJanus. PhunSweet Janus (Appendix A) is similar to Phun Janus, but with the insertion of *sacB* between *pheS* A315G and *kan*, and was generated by SOE PCR (PCR 5) of two fragments with primers HE06/HE07. The upstream fragment (PCR K) included *dexB* plus *pheS* A315G with promoter and was amplified from TIGR4SΔcps::PhunJanus using primers HE06/HE17. The downstream fragment (PCR L) was comprised of *sacB* plus the promoter region upstream of *sacB*, *kan,* and *rpsL* from the Sweet Janus cassette, with an overhang complementary to the 3′ end of *pheS*. This fragment was amplified from SpnYL001 using primers HE21/HE07. TIGR4S and D39S were transformed with PCR 5 to generate TIGR4SΔcps::PhunSweetJanus and D39SΔcps::PhunSweetJanus. PhunSweet (Appendix A) was generated through SOE PCR (PCR 6) of three fragments with primers HE06/HE07. The upstream fragment (PCR M) consisted of *dexB* plus *pheS* A315G with promoter, with an overhang complementary to *sacB* plus promoter. This fragment was amplified from TIGR4SΔcps::PhunJanus using primers HE06/HE22. The middle fragment (PCR N) was comprised of *sacB* plus the promoter region upstream of *sacB* and *kan* and was amplified from SpnYL001 using primers HE10/HE23. The downstream fragment (PCR O) included the segment of *aliA* plus the sequence with D39 homology upstream of *aliA*, with an overhang complementary to the 3′ end of *kan*, and was amplified from SpnYL001 using primers HE24/HE07. PCR 6 was introduced into TIGR4 and D39 to generate TIGR4Δcps::PhunSweet and D39Δcps::PhunSweet. Phun (Appendix A) consisted of *pheS* A315G and *kan* and was generated through SOE PCR (PCR 7) of three fragments with primers HE06/HE07. The upstream (PCR K) and downstream fragments (PCR O) have been described above. The middle fragment (PCR P) included *kan* with an overhang complementary to the 3′ end of *pheS* A315G and was amplified from SpnYL001 with primers HE20/HE23. TIGR4 and D39 were transformed with PCR 7 to generate TIGR4Δcps::Phun and D39Δcps::Phun. All mutants were confirmed to be unencapsulated and sensitive to counter-selection agents (Table 5). Any colony that was not sensitive to the counter-selection agents was not considered to be a mutant. One colony for each mutation was selected to determine allelic replacement efficacy via transformation with 4 µg of D39 gDNA and plating on the appropriate counter-selection agents as described above. The sequence of each cassette is available through NCBI—Phun Janus (#MT684775), PhunSweet Janus (#MT684776), PhunSweet (#MT684774), and Phun (#MT684777).

## 3. Results

### 3.1. Variances in Frequency of False-Positive Transformant Colonies

With its multiple potential applications, we sought to exploit the Sweet Janus system to create capsule swapped variants in our background strain [11]. To maintain an isogenic cell line, we utilized this system to generate an unencapsulated form of our TIGR4 through allelic replacement of the capsule locus with Sweet Janus. Since the Sweet Janus cassette relies upon streptomycin resistance in the background strain, we first generated a streptomycin resistant variant of TIGR4. To reduce the chance of obtaining secondary mutations that could arise from spontaneous mutation, we conferred streptomycin resistance by transformation of TIGR4 with a mutated *rpsL*, which encodes ribosomal protein S12. The single substitution mutation of a lysine to threonine at position 56 (K56T) in *rpsL* confers a high level of streptomycin resistance [21]. We next replaced the capsule locus by transformation of TIGR4S with an amplicon of the entire Sweet Janus locus plus the homologous flanking regions, DexB and AliA, from strain SpnYL001, generously provided by Yuan Li from the Harvard School of Public Health. We had no difficulty obtaining correct transformants for either TIGR4S or for TIGR4SΔcps::SweetJanus. However, after several attempts, we could not obtain a capsule-swapped variant of type4 and a limited number of colonies that recombined type2 capsule despite having hundreds of transformant colonies. To confirm the counter-selection agents were accurate, we concurrently transformed SpnYL001 and TIGR4SΔcps::SweetJanus—both of which replaced the capsule locus with the same cassette—With D39 genomic DNA. As observed previously [11], SpnYL001 readily picked up the type2 capsule, with a low number of false-positives (Figure 1). In contrast, we obtained only seven positive colonies in our TIGR4S::SweetJanus strain but over 500 false-positives, which had the correct growth on the counter-selection agents but did not recombine the type 2 capsule locus. One possible explanation for the low number of true-positives—those colonies that underwent allelic replacement—could be an issue of reduced homology. DexB-SweetJanus-AliA was originally generated by introduction of *sacB* into DexB-Janus-AliA [8,11]. However, these cassettes were generated from the capsule locus of D39 and included regions of chromosome upstream of AliA that are not shared by TIGR4 (Appendix A). To determine if insufficient homology could explain the reduced ability of our TIGR4SΔcps::SweetJanus to recombine the capsule locus, we created a modified DexB-SweetJanus-AliA cassette where Sweet Janus was flanked by the entirety of DexB and AliA from TIGR4—named DexB-NewSweetJanus-AliA. However, transformation of TIGR4SΔcps::NewSweetJanus with D39 genomic DNA still had a low number of true-positives amid a high number of false-positive colonies (Figure 1).

### 3.2. Potential Reversion Necessitates Initial Screening for Mass Application

Since extended homology did not improve the frequency of allelic replacement, we next considered other aspects of this system. Sweet Janus was originally created to enhance counter-selection as streptomycin selection in Janus alone was not stringent enough to reduce the number of potential false-positive colonies. With the addition of *sacB* in Sweet Janus, the frequency of obtaining dual reversion was 10^4^-fold lower than the frequency of *rpsL* reversion alone [11]. Loss of streptomycin sensitivity can occur through spontaneous mutation in *rpsL* or through recombination with the resistant variant of *rpsL* present in the TIGR4S genome, with the latter likely occurring more frequently [7]. Inactivation of *sacB* can occur through spontaneous mutation, requiring as little as a single point mutation of C to T at position 1078 [11,22]. It is possible that, the pneumococcus could gain reversion against either *sacB*, *rpsL*, or both counter-selection agents. This reversion could occur during the first step (generation of TIGR4SΔcps::SweetJanus) or second step (allelic replacement) transformations. A reversion occurring during the first step transformation would become quickly apparent as, upon the second step transformation, a lawn of growth would be present, and isolated colonies would be absent. We observed isolated colonies upon the second step transformation, so it is unlikely reversion occurred during the first step. Indeed, we confirmed that TIGR4SΔcps::SweetJanus and TIGR4SΔcps::NewSweetJanus had no growth on the counter-selection agents (streptomycin plus sucrose). If reversion occurs during the second step transformation, then identification of true-positive colonies amongst false-positive colonies becomes more difficult, and further screening is required. It is possible that there is an inherent reason why a particular bacterial clone would be more likely to revert than another parallel clone.

To test if variance of background resistance depended on the particular bacterial clone, we repeated the transformation steps to generate isolated colonies of TIGR4SΔcps::NewSweetJanus. Since TIGR4 undergoes two transformation steps to generate TIGR4SΔcps::NewSweetJanus, it is possible that variances can occur at each step. To account for this, we isolated eight colonies after transforming *rpsL* K56T to generate TIGR4S, followed by isolation of two colonies of each of these eight TIGR4S colonies after transformation with the NewSweet Janus cassette. All colonies were confirmed to have no background resistance to the counter-selection agents (streptomycin plus sucrose) prior to allelic replacement transformation. Upon transformation with D39 genomic DNA, we observed variances in the number of true-positive (capsule positive) and false-positives (capsule negative) (Figure 2). Some clones (#7 and 8) had an innumerable amount of false-positive colonies with no true-positive colonies. Other clones (#1-1 and 4-1) had a few true-positive colonies, while having high levels of false-positive colonies. Two clones (#2-2 and 5-2) had several true-positive colonies despite high levels of false-positive colonies. One clone (#6-1) stood out in that it had a high number of true-positive colonies but a relatively low number of false-positive colonies. This result suggests that the frequency of reversion in pneumococcus is dependent on the particular bacterial clone. If a one-time allelic replacement is required, then screening the colonies from the second step transformation for loss of kanamycin resistance is a feasible solution to this problem. However, if multiple allelic replacements into the same background is required (e.g., capsule swapping several capsule types into the same genetic background), then screening hundreds of colonies from each of the transformations becomes unreasonable and time-consuming. Thus, if numerous allelic replacements are required, an initial screen to obtain a clone that has an increased true-positive/false-positive ratio (Table 6) is warranted.

To confirm that this initial selection yields a clone that is more efficient at recombining the desired allelic replacement (in this study-capsule), we utilized the TIGR4SΔcps::NewSweetJanus clone that had the highest true-positive/false-positive ratio (#6-1) to create capsule swap variants. While the original Sweet Janus and NewSweet Janus capsule deletion clones in our TIGR4 were ineffective at recombining capsule (Figure 1), TIGR4SΔcps::NewSweetJanus clone #6-1 had a high efficiency at recombining capsule, with correct allelic replacement occurring for seven of 13 capsule types on the first transformation attempt and 3 more on the second transformation attempt (Figure 3a, Table 7). To further confirm that vetting the transformation strain streamlines further allelic replacement, we identified a clone of D39SΔcps::NewSweetJanus that had a high true-positive/false-positive ratio and subjected it to transformation with genomic DNA from 13 different serotypes, with correct allelic replacement occurring for 11 of 13 capsule types (Figure 3b).

To confirm that the observed variances in transformation efficacy was not solely due to the large recombination event required for allelic replacement of the capsule locus, we determined the transformation efficacy of allelic replacement of *spxB* (Sp_0730) in TIGR4S, D39S, as well as, in two clinical backgrounds—ABCA31S and CDC001S. While the total number of true-positive colonies was dramatically increased compared to those of allelic replacement of the capsule locus, we still observed variances in the transformation efficacy depending on the initial clone (Appendix A). This result suggests that, while an initial screen to obtain a clone with increased true-positive/false-positive ratio could reduce downstream applications, the demand to perform the initial screen may depend on the rate of recombination for a particular gene due to its size or to the fitness cost of gene disruption.

### 3.3. Generation of Novel Markerless Cassettes

With the potential for reversion, the pneumococcus can exhibit high levels of false-positive colonies upon transformation (Figure 1 and Figure 2). It is possible that other selection markers may work more effectively than streptomycin and sucrose. A potential alternative selection marker is chlorinated phenylalanine. In streptococci and lactococci, phenylalanyl-tRNA synthetase, encoded by *pheS*, specifically catalyzes attachment of phenylalanine to its cognate tRNA. A point mutation in *pheS* (A315G) renders the synthetase more promiscuous, whereby it becomes capable of attaching a chlorinated analog of phenylalanine. Incorporation of the chlorinated phenylalanine into proteins becomes deleterious and cells with this mutation are unable to grow [20,23]. This selection agent potentially could be used in conjunction with the Janus or Sweet Janus cassette in *S. pneumoniae*, providing a new dual selection marker, chlorinated phenylalanine and streptomycin, or a triple selection marker, chlorinated phenylalanine, streptomycin, and sucrose. Moreover, other combinations of selection markers are possible, including dual selection agents, chlorinated phenylalanine and sucrose, or single-selection agent chlorinated phenylalanine. These latter constructs would have the benefit of not requiring engineered resistance in the background strain since pneumococcus naturally does not integrate chlorinated phenylalanine and, therefore, growth on chlorinated phenylalanine is not inhibited. This unlocks a new set of applications for mutated strains that completely lack any non-native resistances. To test the efficacy of these potential new selection markers, we generated cassettes for deletion of capsule using the different combination of selection markers (Figure 4). Since variability in bacterial clone (rather than differences in homology) impacted allelic replacement efficacy (Figure 1), these new cassettes were designed to incorporate the *dexB* and *aliA* homology from Sweet Janus [11]. For dual chlorinated phenylalanine and streptomycin counter-selection, a mutated *pheS* was introduced upstream of the Janus cassette, replacing the *sacB* of Sweet Janus, and was termed Phun Janus (Appendix A). To ensure proper expression, the promoter incorporated into Sweet Janus was duplicated directly upstream of *pheS*. For triple chlorinated phenylalanine, streptomycin, and sucrose counter-selection, a mutated *pheS* was introduced upstream of the Sweet Janus cassette through splicing of Sweet Janus and Phun Janus and was termed PhunSweet Janus (Appendix A). To generate cassettes that did not rely on conferring the background strain with resistance, we constructed the PhunSweet cassette, which was comprised of a mutated *pheS* followed by *sacB* upstream of kanamycin resistance (Appendix A), and the Phun cassette, which included the mutated *pheS* gene followed by kanamycin resistance (Appendix A).

To confirm these new cassettes function in the pneumococcus, we transformed TIGR4/TIGR4S and D39/D39S with each of these cassettes, which replace the capsule loci through homologous recombination with *dexB* and *aliA*. We isolated colonies that were unencapsulated and had no resistance to the counter-selection agents (Table 5). We then transformed each of the new strains with D39 genomic DNA and screened with the appropriate counter-selection agents. For all strains that contained the mutated *pheS*, we selected transformants on three different concentrations (7.5, 10, and 15 mM) of chlorinated phenylalanine to identify the concentration with the highest stringency [20,23]. We were able to obtain positive recombination of capsule for all strains except for TIGR4Δcps::Phun (Figure 5). Here, 7.5–10 mM chlorinated phenylalanine for TIGR4 and 15 mM chlorinated phenylalanine for D39 yielded the best true-positive to false-positive ratios overall (Table 6). While using chlorinated phenylalanine alone (Phun) as a counter-selection agent is feasible, without a second agent, the frequency of false-positive is much higher than that with two or more counter-selection agents (Figure 5), similar to *sacB* or *rpsL* alone [11].

## 4. Discussion

Possessing molecular tools is invaluable to the investigation of bacterial pathogenesis. The development of Janus, Sweet Janus, and other genome editing methods, such as MuGENT, has aided in studying deletion and allelic replacement of genes in *S. pneumoniae* [7,11,24]. In this study, through screening of sixteen isolated clones, we determined that the efficacy of allelic replacement can vary according to bacterial clone in that the ratio of true-positive/false-positive colonies varied. It is likely that there is an intrinsic element(s) that is altered during cell culture or transformation that leads to variances in parallel isolated colonies. As previously described, the frequency of reversion to streptomycin resistance could be impacted by the *rpsL* allele in the chromosome copy, the presence of RecA, and the activity of HexA [7]. SpnYL001 and TIGR4S streptomycin resistant background strains had the same single point mutation K56T, as confirmed by sequencing. With a single point mutation, the activity of HexA should have limited impact on the reversion of *rpsL* in Sweet Janus [7]. It is possible that the variability observed in revertant colonies could rely on expression or modification of RecA or some other component of recombination, competency, or mismatch repair. The variability could also stem from mutation of *sacB* resulting in loss of sucrose sensitivity. The cause for inconsistency may change upon each transformation and depend on the strain background, warranting a thorough initial screen to identify clones with reduced frequency of reversion. A similar screen was suggested previously as an alternative to alterations to HexA or RecA [7].

One aspect of the Janus and Sweet Janus systems that should be considered is the reliance on introduction of streptomycin resistance into the strain background. In this study, we created novel cassettes that do not require alterations to the strain background (i.e., introduction of resistance) before deletion of the gene of interest and instead rely on altered enzymatic activity to deleteriously incorporate chlorinated phenylalanine into proteins. We have also created cassettes that utilize chlorinated phenylalanine sensitivity in conjunction with those markers found in Sweet Janus. These cassettes are based on the same principal of Janus and follow a two-step transformation procedure (Figure 6). In the first step, transformation of *S. pneumoniae* with the cassette spliced with the flanking regions of the target gene disrupts the gene through homologous recombination and mutants are selected on kanamycin. In the second step, the mutants from the first step are transformed with an alternate allele that replaces the cassette through homologous recombination, and transformants are selected on the appropriate counter-selection agent: chlorinated phenylalanine (Phun), chlorinated phenylalanine and sucrose (PhunSweet); chlorinated phenylalanine and streptomycin (Phun Janus); and chlorinated phenylalanine, streptomycin, and sucrose (PhunSweet Janus). Similar to Sweet Janus, these cassettes demonstrate variable rates of transformation efficacy that may hinge upon selection of the initial clone. Although these cassettes do not overcome the pitfall of potential reversion, they offer new possibilities for molecular manipulation of *S. pneumoniae* and provide a multitude of counter-selection agents for use depending on the restrictions of the experiment.

## 5. Conclusions

In this study, we have identified a potential drawback when utilizing Sweet Janus for large scale allelic replacement in *S. pneumoniae*. We determined that the level of false-positive colonies was dependent on the initial selected clone. Thus, a pre-screening measure may be warranted to identify a clone with a low number of false-positive colonies in order to streamline numerous allelic replacements in the same background. We have also generated several new constructs for allelic replacement that provide alternative selection agents to those in Sweet Janus. Two of the constructs (PhunSweet and Phun) do not rely on conferring resistance to the background strain, providing a means of generating a true markerless genetic replacement. These new constructs provide a new method for genetic manipulation of *S. pneumoniae*, expanding experimental applications.

## Figures and Tables

**Figure 1 genes-11-00965-f001:**
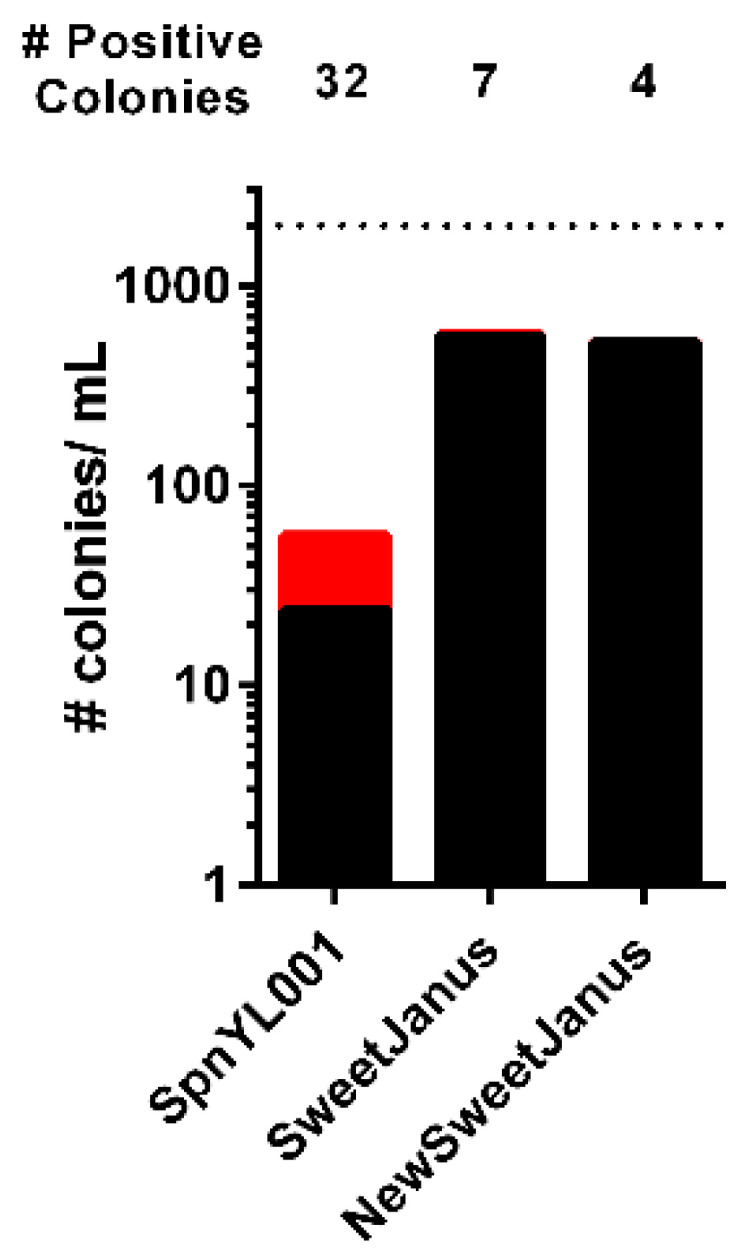
Low efficiency was observed for allelic replacement of Sweet Janus in TIGR4S. SpnYL001, TIGR4SΔcps::SweetJanus, and TIGR4SΔcps::NewSweetJanus were transformed with D39 genomic DNA. The number of true-positive colonies (red bars, number above column) and the number of false-positive colonies (black bars) were determined per mL of transformation. Dashed line represents upper limit of detection.

**Figure 2 genes-11-00965-f002:**
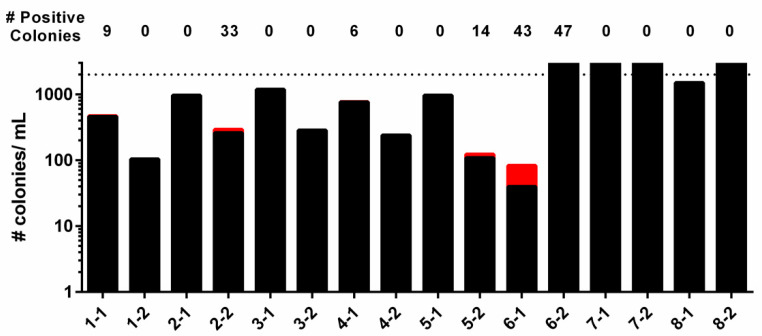
Efficiency of allelic replacement of NewSweet Janus in TIGR4S varied by isolated colony. Eight isolated colonies were selected from the transformation generating TIGR4S, followed by selection of two isolated colonies from each transformation generating TIGR4SΔcps::NewSweetJanus from the eight TIGR4S colonies. The colony numbers are listed as first transformation-second transformation. For example, “1-1” represents the first TIGR4SΔcps::NewSweetJanus colony in TIGR4S clone #1 and “2-1” represents the first TIGR4SΔcps::NewSweetJanus colony in TIGR4S clone #2. The sixteen colonies of TIGR4SΔcps::NewSweetJanus were transformed with D39 genomic DNA. The number of true-positive colonies (red bars, number above column) and the number of false-positive colonies (black bars) were determined per mL of transformation. Dashed line represents upper limit of detection.

**Figure 3 genes-11-00965-f003:**
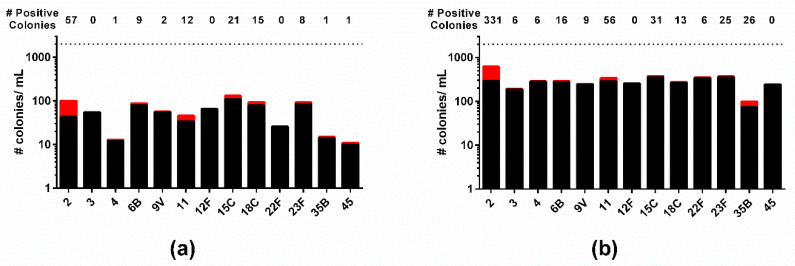
Identification of clones with high efficiency of allelic replacement streamlined multitude of capsule swaps. (**a**) TIGR4SΔcps::NewSweetJanus and (**b**) D39SΔcps::NewSweetJanus were transformed with genomic DNA from strains of different serotypes (below column). The number of true-positive colonies (red bars, number above column) and the number of false-positive colonies (black bars) were determined per mL of transformation. Dashed line represents upper limit of detection.

**Figure 4 genes-11-00965-f004:**
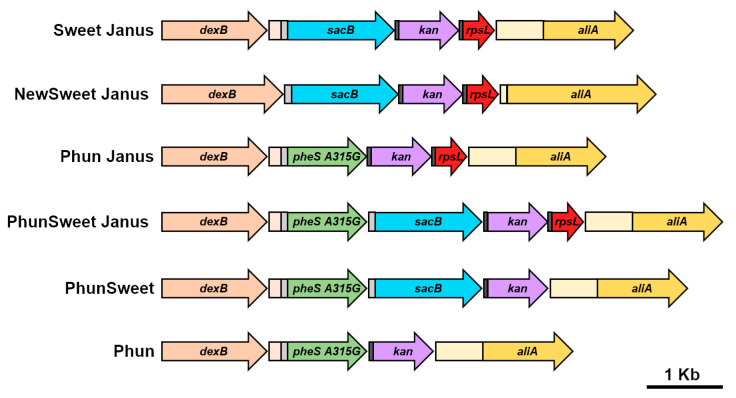
Schematic of novel markerless cassettes generated for allelic replacement of capsule. Each cassette was spliced between the genes flanking the capsule loci, *dexB* (tan) and *aliA* (yellow). The cassettes are comprised of genes that confer sucrose sensitivity (*sacB*; blue), kanamycin resistance (*kan*, purple), streptomycin sensitivity (*rpsL*, red), and chlorinated phenylalanine sensitivity (*pheS* A315G, green). The light gray box represents the promoter from Sweet Janus and the dark gray box represents ribosome-binding site. The light-tan box represents the sequence downstream of *dexB* and the light-yellow box represents the sequence upstream of *aliA* from D39. The direction of the arrow indicates the gene orientation.

**Figure 5 genes-11-00965-f005:**
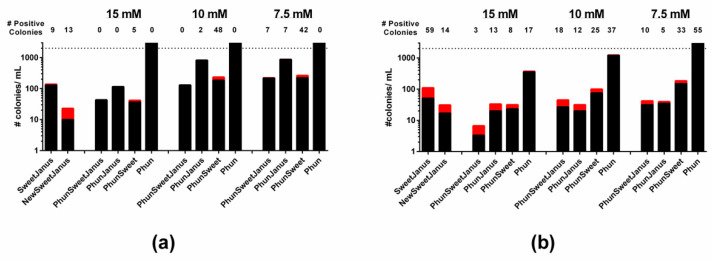
Novel cassettes are effective for allelic replacement of capsule. Cassettes (below column) replaced the capsule loci in (**a**) TIGR4/TIGR4S and (**b**) D39/D39S. The acaspular mutants were transformed with D39 genomic DNA and transformants were selected on counter-selection agents listed in Table 5, with the concentration of chlorinated phenylamine ranging from 7.5 mM to 15 mM. The number of true-positive colonies (red bars, number above column) and the number of false-positive colonies (black bars) were determined per mL of transformation. Dashed line represents upper limit of detection.

**Figure 6 genes-11-00965-f006:**
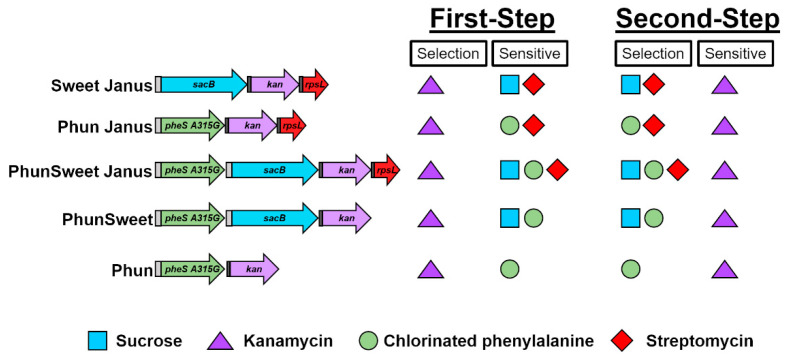
Mutation of *S. pneumoniae* using novel cassettes occurs in a two-step transformation procedure. The schematic indicates the selection agents and the agents that inhibit growth of transformants for each transformation step. The selection agents are represented by a colored shape: blue square for sucrose; purple triangle for kanamycin; green circle for chlorinated phenylalanine; red diamond for streptomycin.

**Table 1 genes-11-00965-t001:** Strains used in this study.

Strain	Description	Source
SPNYL001	Capsule locus replaced with Sweet Janus	Li, et al. [11]
TIGR4	TIGR4 wild type, serotype 4	
TIGR4S	TIGR4 with K56T mutation in *rpsL* (Sp_0271)	This study
TIGR4SΔcps::SweetJanus	Capsule locus replaced with Sweet Janus	This study
TIGR4SΔcps::NewSweetJanus	Capsule locus replaced with NewSweet Janus	This study
TIGR4SΔcps::PhunJanus	Capsule locus replaced with Phun Janus	This study
TIGR4SΔcps::PhunSweetJanus	Capsule locus replaced with PhunSweet Janus	This study
TIGR4Δcps::PhunSweet	Capsule locus replaced with PhunSweet	This study
TIGR4Δcps::Phun	Capsule locus replaced with Phun	This study
D39	D39 wild type, serotype 2	
D39S	D39 with K56T mutation in *rpsL* (Sp_0251)	This study
D39SΔcps::SweetJanus	Capsule locus replaced with Sweet Janus	This study
D39SΔcps::NewSweetJanus	Capsule locus replaced with NewSweet Janus	This study
D39SΔcps::PhunJanus	Capsule locus replaced with Phun Janus	This study
D39SΔcps::PhunSweetJanus	Capsule locus replaced with PhunSweet Janus	This study
D39Δcps::PhunSweet	Capsule locus replaced with PhunSweet	This study
D39Δcps::Phun	Capsule locus replaced with Phun	This study
ABCA38	Serotype 22F	CDC
CDC007	Serotype 6B	CDC
ABCA31	Serotype 35B	CDC
45	Serotype 45	CDC
CDC001	Serotype 9V	CDC
CDC004	Serotype 23F	CDC
DAW4	Serotype 11	CDC
ABCA61	Serotype 3	CDC
CDC030	Serotype 12F	CDC
CDC048	Serotype 18C	CDC
CDC009	Serotype 15C	CDC
TIGR4ΔspxB::Erm	*spxB* (Sp_0730) replaced with Erm	Echlin, et al. [12]
TIGR4SΔspxB::SweetJanus	*spxB* (Sp_0730) replaced with Sweet Janus	This study
D39SΔspxB::SweetJanus	*spxB* (Spd_0636) replaced with Sweet Janus	This study
ABCA31SΔspxB::SweetJanus	*spxB* replaced with Sweet Janus	This study
CDC001SΔspxB::SweetJanus	*spxB* replaced with Sweet Janus	This study

**Table 2 genes-11-00965-t002:** Strain resistance (R) and susceptibility (S) to selection agents: 800 µg/mL streptomycin (Strep), 400 µg/mL kanamycin (Kan), 10% sucrose (Suc), and 10 mM chlorinated-phenylalanine (Chl-Phe).

Strain	Strep	Kan	Suc	Chl-Phe
SpnYL001	S	R	S	R
TIGR4	S	S	R	R
TIGR4S	R	S	R	R
TIGR4SΔcps::SweetJanus	S	R	S	R
TIGR4SΔcps::NewSweetJanus	S	R	S	R
TIGR4SΔcps::PhunJanus	S	R	R	S
TIGR4SΔcps::PhunSweetJanus	S	R	S	S
TIGR4Δcps::PhunSweet	S	R	S	S
TIGR4Δcps::Phun	S	R	R	S
D39	S	S	R	R
D39S	R	S	R	R
D39SΔcps::SweetJanus	S	R	S	R
D39SΔcps::NewSweetJanus	S	R	S	R
D39SΔcps::PhunJanus	S	R	R	S
D39SΔcps::PhunSweetJanus	S	R	S	S
D39Δcps::PhunSweet	S	R	S	S
D39Δcps::Phun	S	R	R	S

**Table 3 genes-11-00965-t003:** Primers used in this study. Point mutations are underlined.

Primer	Sequence
HE01	GCCGTAGTCATCTTTCTTGGCATC
HE02	CTGAGTTAGGTTTTGTAGGTGTCATTGTTC
HE03	GAACAATGACACCTACAAAACCTAACTCAG
HE04	CTAATTTGAACCCGGGCTAAAGTTAG
HE05	CTGTCACCAAGTGTATCATCACC
HE06	GATTGCGGCTATTTTTGGAACCATGG
HE07	CATCCTTCCATTCATCCCCATAAGTGAC
HE08	ATGCAAGAAAAATGGTGGCACAATG
HE09	ATTAAAAATCAAACGGATCGATCCTTAA-TTATAGTAATTCCACACAGAAAGCATCCCATG
HE10	TTAAGGATCGATCCGTTTGATTTTTAATGG
HE11	TTATGCTTTTGGACGTTTAGTACCGTATTTAG
HE12	CGGTACTAAACGTCCAAAAGCATAA-GAATGATAGATACCTTGTTATGACGCGCTTAC
HE13	TTATTTCACATGTTTTGCGAGATCTTCTTG
HE14	ATGTCAACTATTGAAGAACAATTAAAAGCG
HE15	GTCCAAGACCAAAGCCAAAGCCAGAGTATAC
HE16	GTATACTCTGGCTTTGGCTTTGGTCTTGGAC
HE17	TTATTTAAACTGTTCTGAGAAGCGGACATC
HE18	GTGTGGTGGAGAAGGCTGTAATGTATG
HE19	CGCTTTTAATTGTTCTTCAATAGTTGACAT-TTATTATTTCCCTCCTCTTTTCTACAG
HE20	GATGTCCGCTTCTCAGAACAGTTTAAATAA-AAACGCAAAAGAAAATGCCGATGGCCGCCC
HE21	GATGTCCGCTTCTCAGAACAGTTTAAATAA-TTAAGGATCGATCCGTTTGATTTTTAATGG
HE22	CCATTAAAAATCAAACGGATCGATCCTTAA-TTATTTAAACTGTTCTGAGAAGCGGACATC
HE23	CTAAAACAATTCATCCAGTAAAAT
HE24	ATTTTACTGGATGAATTGTTTTAG-GCCGGCGTCAGTCAGTTTT
HE25	GCGAGCGAGTGAAGCTGG
HE26	TCAAACGGATCGATCCTTAA-AATGATAACTCTCCTTCAATTTTTTTAAACTTGGAG
HE27	ACTAAACGTCCAAAAGCATAA-TTCCTCTCGCCGAAAATCAAATATGAAACTTG
HE28	CCATAGTCACTATATACGAGAATTTCGC

**Table 4 genes-11-00965-t004:** PCR (polymerase chain reaction) setup for cassette. Words in parentheses indicate overhang homology.

PCR	Product	F Primer	R Primer	gDNA Template	Size
A	RpsL_K56T Up	HE01	HE02	TIGR4	1149
B	RpsL_K56T Down	HE03	HE04	TIGR4	1015
C	DexB-SweetJanus-AliA	HE06	HE07	SpnYL001	6180
D	DexB (SacB)	HE08	HE09	TIGR4	1608
E	SacB-Kan-RpsL	HE10	HE11	SpnYL001	2807
F	AliA (RpsL)	HE12	HE13	TIGR4	2135
G	PheS A315G Up	HE14	HE15	TIGR4	958
H	PheS A315G Down	HE16	HE17	TIGR4	120
I	DexB + Prom (PheS)	HE06	HE19	SpnYL001	1622
J	Kan-RpsL-AliA(PheS)	HE20	HE07	SpnYL001	3135
K	DexB-PheS A315G	HE06	HE17	TIGR4SΔcps::PhunJanus	2669
L	SacB-Kan-RpsL-AliA (PheS)	HE21	HE07	SpnYL001	4638
M	DexB-PheS A315G (SacB)	HE06	HE22	TIGR4SΔcps::PhunJanus	2669
N	SacB-Kan	HE10	HE23	SpnYL001	2350
O	AliA	HE24	HE07	SpnYL001	1819
P	SJ Kan	HE20	HE23	SpnYL001	847
**SOE**	**Product**	**F Primer**	**R Primer**	**PCR Parts**	**Size**
1	RpsL_K56T	HE01	HE04	A,B	2133
2	DexB-NewSweetJanus-AliA	HE08	HE13	D,E,F	6550
3	PheS A315G	HE14	HE17	G,H	1047
4	DexB-PhunJanus-AliA	HE06	HE07	I,J,3	5804
5	DexB-PhunSweetJanus-AliA	HE06	HE07	K,L	7307
6	DexB-PhunSweet-AliA	HE06	HE07	M,N,O	6838
7	DexB-Phun-AliA	HE06	HE07	K,O,P	5335

**Table 5 genes-11-00965-t005:** Selective agents used in TSA (tryptic soy agar) plates when replacing the markerless cassette in step two transformation.

Markerless Cassette	Counter-Selection Agents
Sweet Janus	800 µg/mL Streptomycin + 10% Sucrose
NewSweet Janus	800 µg/mL Streptomycin + 10% Sucrose
Phun Janus	800 µg/mL Streptomycin + 7.5–15 mM Chlorinated-Phenylalanine
PhunSweet Janus	800 µg/mL Streptomycin + 10% Sucrose + 7.5–15 mM Chlorinated-Phenylalanine
PhunSweet	10% Sucrose + 7.5–15 mM Chlorinated-Phenylalanine
Phun	7.5–15 mM Chlorinated-Phenylalanine

**Table 6 genes-11-00965-t006:** Ratio of true positive to false positive colonies. Selection occurred on 800 µg/mL streptomycin (Strep), 10% sucrose (Suc), or 15–7.5mM chlorinated-phenylalanine (Chl-Phe).

Strain	Cassette	Selection	Ratio
SPNY001	SweetJanus	Strep, Suc	0.311
TIGR4S	SweetJanus	Strep, Suc	0.014
TIGR4S	NewSweetJanus	Strep, Suc	0.009
TIGR4S, Clone 1	NewSweetJanus, Clone 1	Strep, Suc	0.019
TIGR4S, Clone 1	NewSweetJanus, Clone 2	Strep, Suc	0
TIGR4S, Clone 2	NewSweetJanus, Clone 1	Strep, Suc	0
TIGR4S, Clone 2	NewSweetJanus, Clone 2	Strep, Suc	0.13
TIGR4S, Clone 3	NewSweetJanus, Clone 1	Strep, Suc	0.000
TIGR4S, Clone 3	NewSweetJanus, Clone 2	Strep, Suc	0
TIGR4S, Clone 4	NewSweetJanus, Clone 1	Strep, Suc	0.008
TIGR4S, Clone 4	NewSweetJanus, Clone 2	Strep, Suc	0
TIGR4S, Clone 5	NewSweetJanus, Clone 1	Strep, Suc	0
TIGR4S, Clone 5	NewSweetJanus, Clone 2	Strep, Suc	0.13
TIGR4S, Clone 6	NewSweetJanus, Clone 1	Strep, Suc	1.075
TIGR4S, Clone 6	NewSweetJanus, Clone 2	Strep, Suc	<0.02
TIGR4S, Clone 7	NewSweetJanus, Clone 1	Strep, Suc	0
TIGR4S, Clone 7	NewSweetJanus, Clone 2	Strep, Suc	0
TIGR4S, Clone 8	NewSweetJanus, Clone 1	Strep, Suc	0
TIGR4S, Clone 8	NewSweetJanus, Clone 2	Strep, Suc	0
TIGR4S	PhunJanus	Strep, Chl-Phe15mM	0
TIGR4S	PhunJanus	Strep, Chl-Phe10mM	0.002
TIGR4S	PhunJanus	Strep, Chl-Phe7.5mM	0.008
TIGR4S	PhunSweetJanus	Strep, Suc, Chl-Phe15mM	0
TIGR4S	PhunSweetJanus	Strep, Suc, Chl-Phe10mM	0
TIGR4S	PhunSweetJanus	Strep, Suc, Chl-Phe7.5mM	0.031
TIGR4	PhunSweet	Suc, Chl-Phe15mM	0.136
TIGR4	PhunSweet	Suc, Chl-Phe10mM	0.261
TIGR4	PhunSweet	Suc, Chl-Phe7.5mM	0.187
TIGR4	Phun	Chl-Phe15mM	0
TIGR4	Phun	Chl-Phe10mM	0
TIGR4	Phun	Chl-Phe7.5mM	0
D39S	PhunJanus	Strep, Chl-Phe15mM	0.667
D39S	PhunJanus	Strep, Chl-Phe10mM	0.583
D39S	PhunJanus	Strep, Chl-Phe7.5mM	0.143
D39	PhunSweetJanus	Strep, Suc, Chl-Phe15mM	1.000
D39	PhunSweetJanus	Strep, Suc, Chl-Phe10mM	0.688
D39	PhunSweetJanus	Strep, Suc, Chl-Phe7.5mM	0.316
D39	PhunSweet	Suc, Chl-Phe15mM	0.357
D39	PhunSweet	Suc, Chl-Phe10mM	0.333
D39	PhunSweet	Suc, Chl-Phe7.5mM	0.233
D39	Phun	Chl-Phe15mM	0.047
D39	Phun	Chl-Phe10mM	0.030
D39	Phun	Chl-Phe7.5mM	0.019

**Table 7 genes-11-00965-t007:** Ratio of true positive to false positive colonies of TIGR4SΔcps::NewSweetJanus and D39SΔcps::NewSweetJanus transformed with genomic DNA of different serotypes. NSJ refers to NewSweetJanus. Selection occurred on 800 µg/mL streptomycin (Strep), 10% sucrose (Suc), or 7.5–15 mM chlorinated-phenylalanine (Chl-Phe).

Strain	gDNA	Selection	Ratio
TIGR4SΔcps::NSJ	TIGR4 (4)	Strep, Suc	0.08
TIGR4SΔcps::NSJ	D39 (2)	Strep, Suc	1.33
TIGR4SΔcps::NSJ	ABCA38 (22F)	Strep, Suc	0
TIGR4SΔcps::NSJ	CDC007 (6B)	Strep, Suc	0.11
TIGR4SΔcps::NSJ	ABCA31 (35B)	Strep, Suc	0.07
TIGR4SΔcps::NSJ	45	Strep, Suc	0.10
TIGR4SΔcps::NSJ	CDC001 (9V)	Strep, Suc	0.04
TIGR4SΔcps::NSJ	CDC004 (23F)	Strep, Suc	0.09
TIGR4SΔcps::NSJ	DAW4 (11)	Strep, Suc	0.35
TIGR4SΔcps::NSJ	ABCA61 (3)	Strep, Suc	0
TIGR4SΔcps::NSJ	CDC030 (12E)	Strep, Suc	0
TIGR4SΔcps::NSJ	CDC048 (18C)	Strep, Suc	0.19
TIGR4SΔcps::NSJ	CDC009 (15C)	Strep, Suc	0.19
D39SΔcps::NSJ	TIGR4 (4)	Strep, Suc	0.02
D39SΔcps::NSJ	D39 (2)	Strep, Suc	1.14
D39SΔcps::NSJ	ABCA38 (22F)	Strep, Suc	0.02
D39SΔcps::NSJ	CDC007 (6B)	Strep, Suc	0.06
D39SΔcps::NSJ	ABCA31 (35B)	Strep, Suc	0.36
D39SΔcps::NSJ	45	Strep, Suc	0
D39SΔcps::NSJ	CDC001 (9V)	Strep, Suc	0.04
D39SΔcps::NSJ	CDC004 (23F)	Strep, Suc	0.07
D39SΔcps::NSJ	DAW4 (11)	Strep, Suc	0.20
D39SΔcps::NSJ	ABCA61 (3)	Strep, Suc	0.03
D39SΔcps::NSJ	CDC030 (12E)	Strep, Suc	0
D39SΔcps::NSJ	CDC048 (18C)	Strep, Suc	0.05
D39SΔcps::NSJ	CDC009 (15C)	Strep, Suc	0.09

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
