# Peer review of "Advancing Genetic Tools in Streptococcus pneumoniae"

_genes, 2020, doi:10.3390/genes11090965_

Round 1

Reviewer 1 Report

In this manuscript, Haley and Jason described novel genetic tools by introducing a new counter-selection marker (pheS A315G) into Sweet Janus cassette. This new selection marker provides tools that does not require engineered resistance in the background strain. This new cassette will definitely give more options to scientists working in this filled to manipulate Streptococcus genes. While there are few concerns that need to be corrected. 

While this new selection marker provides an additional option to scientists working in this filed, there was no significant increase in the number of positive colonies over false-positive hits compared to NewSweetJauns cassette. Especially, Fig 5a, NewSweetJanus works better than PhunsweetJanus. Considering this, I would suggest the authors re-write their results to clearly present their findings. In short, it will be better to emphasize expanding cassette options than developing tools that work better than SweetJanus. 

Minor Comments.

line 286 should be 3.2 not 3.1

Author Response

RESPONSE TO REVIEWERS

Reviewer 1

In this manuscript, Haley and Jason described novel genetic tools by introducing a new counter-selection marker (pheS A315G) into Sweet Janus cassette. This new selection marker provides tools that does not require engineered resistance in the background strain. This new cassette will definitely give more options to scientists working in this filled to manipulate Streptococcus genes. While there are few concerns that need to be corrected.

COMMENT: While this new selection marker provides an additional option to scientists working in this filed, there was no significant increase in the number of positive colonies over false-positive hits compared to NewSweetJauns cassette. Especially, Fig 5a, NewSweetJanus works better than PhunsweetJanus. Considering this, I would suggest the authors re-write their results to clearly present their findings. In short, it will be better to emphasize expanding cassette options than developing tools that work better than SweetJanus.

RESPONSE: We understand the reviewer’s concern and it was not our intention to suggest that these constructs were more efficient than SweetJanus, rather that they provide a new alternative to circumvent some of SweetJanus restrictions. We have updated our wording to reflect this on lines 68,73-74, 496-499.

COMMENT: Minor Comments.

line 286 should be 3.2 not 3.1

RESPONSE: We have corrected this error, now on line 315.

Reviewer 2 Report

Summary

The manuscript entitled “Advancing Genetic Tools in Streptococcus pneumoniae” presents a thorough evaluation of the current method for markerless genomic modification in S. pneumoniae and propose a novel addition hence expanding the current toolbox for genetic manipulation of this important pathogen.

General assessment

The overall manuscript is well written, concise and appropriately structured. The generation of novel markerless cassettes represents the core of the manuscript and will unquestionably be of great interest for researchers looking to streamline genomic manipulation of pneumococci. However, even though the development of such a tool is greatly welcomed, the presented results raise some questions that should be addressed or discussed. The list below draws attention to some these aspects that should be evaluated:

Major points:

  1. Authors describe a variable efficiency of transformation using Sweet Janus cassette in the TIGR4SΔcps::SweetJanus background. While they did test the hypothesis that imperfect homology might be a cause for low number of true positives, they did not attempt any additional experiments, nor did they try to further explain or propose new hypothesis of why this might be the case. For example, all presented transformation experiments are done with genomic DNA from various pneumococcal strains with the associated readout of rough/smooth colony morphology. Would the result be the same with another loci? I would suggest to authors to evaluate the transformation efficiency with a specific PCR-amplified product targeting another region in the chromosome. For example, authors could design a Sweet Janus cassette targeting a locus outside of the capsule operon and try the second round of transformation with a PCR-amplified product (chloramphenicol, erythromycin or a fluorescent reporter for easy readout). If possible, this experiment should be carried out in few different strains from distinct serotypes (possibly CDC isolates listed in Table 1). Targeting a conserved genomic region/gene would allow to design a unique construct for all strains.

  2. Janus and Sweet Janus have been used extensively to generate multiple markerless genomic modifications. Did other studies report this level of false positives? It is an important point to address as readers might be left with the impression that Sweet Janus cassette is less than optimal way to achieve modification of multiple loci. The high number of false-positive following a second step of transformation might be a general feature of Sweet Janus, however it is also possible that this is either strain-specific or locus specific. The former should be addressed through a careful review of literature, presented in the discussion and the later with my previous comment.

  1. The authors do not offer any explanation why different “cell lines” produce drastically different transformation efficiencies. While I recognize that this aspect is quite puzzling, it suggests a possible heterogeneity in bacterial core processes such as transformation, recombination and possibly mutation. I encourage the authors to go beyond general statements such as: “It is possible that there is an inherent reason why a particular bacterial cell line would be more likely to revert than another parallel line” which offers no food for thought for the reader. I would also suggest the use of the word “clone” instead of “cell line”.

  1. The transformation efficiencies reported in the Figure 3 vary considerable among different serotypes. This could be due to the sequence divergence of the regions flanking the Sweet Janus and genomic locus. This aspect prevents an accurate comparison of transformation efficiencies. For example, for serotypes 3, 12F and 22F for which no positive clones could be isolated (Figure 3A), is it down to an extreme sequence divergence of the supposedly homologous regions flanking the capsule/Sweet Janus cassette? Once again, I believe that this aspect would be best addressed by targeting a highly conserved region in different strains (see comment 1).

  2. I truly admire the great effort by the authors to generate and improve the current method by generating multiple counter-selection systems, however one can’t but feel that transformation ratios (as defined by the ratio of positives vs false-positives) is still low and somewhat comparable to that of Sweet Janus (comparing the ratios in Table 7 and results from the Figure 3). This observation likely points to a different issue with the system and not merely based on revertants (i.e. very unlikely that a high rate of revertants would occur with triple negative selection markers). I would love to see the authors acknowledge and discuss this aspect openly.

  3. The authors should acknowledge the existence of another method for markerless genomic alterations in competent species including pneumoniae described in the paper by Dalia B.A. et al., Multiplex genome editing by natural transformation, PNAS, 2014. A comparison of transformation efficiencies would be valuable.

  4. The authors should avoid using genomic DNA for any transformation due to the high possibility of accumulation of unselected off-target mutations. Transforming a strain using genomic DNA from a different serotype and selecting for capsule will likely introduce a plethora of unwanted mutations, deletions and genomic rearrangements as multiple DNA segments beside the capsule loci are likely acquired at the same time by any given competent cell. Any use of gDNA for transformation should be justified and the resulting strains whole-genome sequenced to confirm the nature of off-target mutations.

Minor points:

  1. The authors base their ration by visual counting of rough vs smooth colonies. However, from my personal experience with pneumoniae, it is rather hard to distinguish those two morphotypes as colonies are rather small (except for serotype 3, which produces large mucoid colonies). Could the authors try another method? Latex agglutionation is labor intensive and can be applied to small number of clones. Perhaps colony PCR a subset (e.g. 50) of clones for each transformation. Once again, introducing a selectable marker or a fluorescent reporter might greatly simplify counting and add a greater certainty that the correct phenotype is being reported.

  2. Line 28: the statement that the current vaccine provides “some protection against this organism” is misleading. The correct statement should imply that the current vaccine has narrow coverage but provides great protection against colonization and invasive disease. Please add recent references.

  3. Line 77: it is unclear why the authors use 20 μg/ ml neomycin for plating their frozen stocks. There is no need for a selection at this point unless their glycerol stocks have contaminants. Could the authors clarify this point in the methodology section?

  4. Line 354: Figure 3 reports counts and as such should be presented on a linear Y-axis. The log scale is very confusing and inappropriate for this type of data.

Author Response

Reviewer 2

Summary

The manuscript entitled “Advancing Genetic Tools in Streptococcus pneumoniae” presents a thorough evaluation of the current method for markerless genomic modification in S. pneumoniae and propose a novel addition hence expanding the current toolbox for genetic manipulation of this important pathogen.

General assessment

The overall manuscript is well written, concise and appropriately structured. The generation of novel markerless cassettes represents the core of the manuscript and will unquestionably be of great interest for researchers looking to streamline genomic manipulation of pneumococci. However, even though the development of such a tool is greatly welcomed, the presented results raise some questions that should be addressed or discussed. The list below draws attention to some these aspects that should be evaluated:

Major points:

COMMENT: Authors describe a variable efficiency of transformation using Sweet Janus cassette in the TIGR4SΔcps::SweetJanus background. While they did test the hypothesis that imperfect homology might be a cause for low number of true positives, they did not attempt any additional experiments, nor did they try to further explain or propose new hypothesis of why this might be the case. For example, all presented transformation experiments are done with genomic DNA from various pneumococcal strains with the associated readout of rough/smooth colony morphology. Would the result be the same with another loci? I would suggest to authors to evaluate the transformation efficiency with a specific PCR-amplified product targeting another region in the chromosome. For example, authors could design a Sweet Janus cassette targeting a locus outside of the capsule operon and try the second round of transformation with a PCR-amplified product (chloramphenicol, erythromycin or a fluorescent reporter for easy readout). If possible, this experiment should be carried out in few different strains from distinct serotypes (possibly CDC isolates listed in Table 1). Targeting a conserved genomic region/gene would allow to design a unique construct for all strains.

RESPONSE: We thank the reviewer for this suggested experiment. We have determined transformation efficacy using another locus- namely spxB. We generated eight colonies of  ΔspxB::SweetJanus in four strain backgrounds—TIGR4, D39, ABCA31, and CDC001. To determine the number of true-positive and false-positive colonies, we replaced the Sweet Janus cassette with a cassette conferring erythromycin resistance and determined the number of colonies on plates with streptomycin plus sucrose or with erythromycin. We observed a variable transformation efficacy depending on the initial colony similar to that observed for allelic replacement of the capsule locus, although the total number of true-positive colonies was higher overall.  This new data is presented in Table S1 and discussed in lines 394-402. Methods for this experiment are listed in lines 188-215, with strains and primers generated listed in Table 1 and Table 3.

COMMENT: Janus and Sweet Janus have been used extensively to generate multiple markerless genomic modifications. Did other studies report this level of false positives? It is an important point to address as readers might be left with the impression that Sweet Janus cassette is less than optimal way to achieve modification of multiple loci. The high number of false-positive following a second step of transformation might be a general feature of Sweet Janus, however it is also possible that this is either strain-specific or locus specific. The former should be addressed through a careful review of literature, presented in the discussion and the later with my previous comment.

RESPONSE: To the authors’ knowledge, no other studies have reported data demonstrating the variability in transformation efficacy using the Janus cassette. In the manuscript in which Janus was generated (Sung et. al.), the authors mention potential differences in the rate of reversion with parallel cell lines: “Interestingly, the Smr revertants may be enriched during colonial culture growth; pure Sms populations of CP1296 grown in liquid culture exhibited a lower frequency of revertants than parallel colonial cultures (data not shown), offering a possible alternative to the Hex-dependent strategy for reducing the rate of gene conversion.” We have cited this on lines 481-482. With the new data presented in Table S1, we have noted that the demand to perform an initial screen may be gene-dependent due to gene size or the fitness cost of gene disruption. Thus, it is possible that an initial screen was not warranted for many of the studies that utilized Sweet Janus as the number of true-positive colonies was high. However, this may not be true for future studies that wish to utilize these systems for loci that are larger or have a greater fitness impact.

COMMENT: The authors do not offer any explanation why different “cell lines” produce drastically different transformation efficiencies. While I recognize that this aspect is quite puzzling, it suggests a possible heterogeneity in bacterial core processes such as transformation, recombination and possibly mutation. I encourage the authors to go beyond general statements such as: “It is possible that there is an inherent reason why a particular bacterial cell line would be more likely to revert than another parallel line” which offers no food for thought for the reader. I would also suggest the use of the word “clone” instead of “cell line”.

RESPONSE: We understand the reviewer’s concern. We have stated potential reasons why different clones have variable transformation efficiencies in the Discussion on lines 471-480, including alterations in RecA (recombination) or HexA (mismatch repair), and competency (transformation) We apologize if this was not clear. We have adjusted our wording from “cell line” to “clone” on lines 15, 61, 335, 336, 351, 422, and 469.

COMMENT: The transformation efficiencies reported in the Figure 3 vary considerable among different serotypes. This could be due to the sequence divergence of the regions flanking the Sweet Janus and genomic locus. This aspect prevents an accurate comparison of transformation efficiencies. For example, for serotypes 3, 12F and 22F for which no positive clones could be isolated (Figure 3A), is it down to an extreme sequence divergence of the supposedly homologous regions flanking the capsule/Sweet Janus cassette? Once again, I believe that this aspect would be best addressed by targeting a highly conserved region in different strains (see comment 1).

RESPONSE: We understand the reviewer’s concern. Although many capsule loci share homology of the dexB and aliA genes, it is possible that the sequence divergences reduce the rate of recombination. However, the purpose of this experiment was to demonstrate that pre-screening for clones with a high true-positive/ false-positive ratio could streamline downstream applications (lines 371-372, and 378-379), rather than to compare efficiency of uptake of different capsule loci. As suggested, we have determined the variability in transformation efficiency in a conserved region (spxB- see above).

*Note*The author’s noted a typographical error for strain CDC030 (should read 12F and not 12E) and have adjusted Table 1 and Figure 3.

COMMENT: I truly admire the great effort by the authors to generate and improve the current method by generating multiple counter-selection systems, however one can’t but feel that transformation ratios (as defined by the ratio of positives vs false-positives) is still low and somewhat comparable to that of Sweet Janus (comparing the ratios in Table 7 and results from the Figure 3). This observation likely points to a different issue with the system and not merely based on revertants (i.e. very unlikely that a high rate of revertants would occur with triple negative selection markers). I would love to see the authors acknowledge and discuss this aspect openly.

RESPONSE: We thank the reviewer and have updated our wording to discuss that these new cassettes provide an alternative to Sweet Janus, but still have a potential for high false-positive colonies, on lines 68,73-74, 496-499.

COMMENT: The authors should acknowledge the existence of another method for markerless genomic alterations in competent species including pneumoniae described in the paper by Dalia B.A. et al., Multiplex genome editing by natural transformation, PNAS, 2014. A comparison of transformation efficiencies would be valuable.

RESPONSE: We have acknowledged the alternative method on lines 466-467. We feel that generation of capsule deletions using this alternative method and comparison of the transformation efficiencies is out of the scope of this study.

COMMENT: The authors should avoid using genomic DNA for any transformation due to the high possibility of accumulation of unselected off-target mutations. Transforming a strain using genomic DNA from a different serotype and selecting for capsule will likely introduce a plethora of unwanted mutations, deletions and genomic rearrangements as multiple DNA segments beside the capsule loci are likely acquired at the same time by any given competent cell. Any use of gDNA for transformation should be justified and the resulting strains whole-genome sequenced to confirm the nature of off-target mutations.

RESPONSE: We understand the reviewer’s concern. For mutations of individual genes, we typically transform S. pneumoniae with SOE PCR fragments. However, in the case of capsule, it is difficult to obtain or obtain without mutations PCR fragments of large pieces of DNA (~ 22kB for the capsule loci). Any further downstream experiments using capsule-swapped strains would be subject to whole-genome sequencing to confirm any off-target mutations. Previous manuscripts have utilized genomic DNA for capsule swaps as well (Li et al, Trzcin´ski et al) and we wanted to utilize the same methodologies that have previously been utilized for such swaps.

Minor points:

COMMENT: The authors base their ration by visual counting of rough vs smooth colonies. However, from my personal experience with pneumoniae, it is rather hard to distinguish those two morphotypes as colonies are rather small (except for serotype 3, which produces large mucoid colonies). Could the authors try another method? Latex agglutionation is labor intensive and can be applied to small number of clones. Perhaps colony PCR a subset (e.g. 50) of clones for each transformation. Once again, introducing a selectable marker or a fluorescent reporter might greatly simplify counting and add a greater certainty that the correct phenotype is being reported.

RESPONSE: We understand the reviewer’s concern. The acapsular version of TIGR4 and D39 are much smaller and non-mucoid compared to their capsular counterparts, particularly when expressing type 2 capsule. To confirm that we had appropriately ascertained the acapsular colonies from the more mucoid capsular colonies, we performed latex agglutination on a subset of each transformation, see lines 174-175.

COMMENT: Line 28: the statement that the current vaccine provides “some protection against this organism” is misleading. The correct statement should imply that the current vaccine has narrow coverage but provides great protection against colonization and invasive disease. Please add recent references.

RESPONSE: We have updated our wording and added references on lines 28-31.

COMMENT: Line 77: it is unclear why the authors use 20 μg/ ml neomycin for plating their frozen stocks. There is no need for a selection at this point unless their glycerol stocks have contaminants. Could the authors clarify this point in the methodology section?

RESPONSE: We understand the reviewer’s concern. We add neomycin as this is standard practice for culturing S. pneumoniae from infected tissues and as such we utilize for all culturing in the lab. We have checked the stocks of strains utilized in this study on plates lacking neomycin to ensure stocks are not contaminated.

COMMENT: Line 354: Figure 3 reports counts and as such should be presented on a linear Y-axis. The log scale is very confusing and inappropriate for this type of data.

RESPONSE: We understand the reviewer’s concern. We have chosen to depict Figures 1-3 in log scale because the red bars are near-invisible due to the large number of false-positive colonies (black bar) in linear scale. To further address this concern we have provided the actual numerical values of the true-positive colonies, listed above each column, and have provided the ratio (true-positive/ false-positive) of each transformation in Tables 6-7.